# Chronological Changes in the Histology of Infection-Related Glomerulonephritis in Renal Allograft: A Case Report

**DOI:** 10.3390/ijms25105095

**Published:** 2024-05-07

**Authors:** Kenta Tominaga, Takashi Oda, Sachiko Iwama, Tadasu Kojima, Osamu Konno, Muneharu Yamada, Iwao Nakabayashi, Hitoshi Iwamoto

**Affiliations:** 1Department of Nephrology, Self-Defense Forces Central Hospital, Setagaya 154-8532, Japan; 2Department of Nephrology and Blood Purification, Kidney Disease Center, Tokyo Medical University Hachioji Medical Center, Hachioji 193-0998, Japan; 3Department of Kidney Transplantation Surgery, Kidney Disease Center, Tokyo Medical University Hachiouji Medical Center, Hachioji 193-0998, Japan; 4Department of Nephrology, Fussa Hospital, Fussa 197-8511, Japan

**Keywords:** infection-related glomerulonephritis, renal allograft, nephritis-associated plasmin receptor, plasmin, interstitial fibrosis and tubular atrophy

## Abstract

We report the histological changes over time for a patient with infection-related glomerulonephritis (IRGN) that developed in a transplanted kidney. A 47-year-old man had undergone renal transplantation 3 years ago for end-stage kidney disease (ESKD). After several episodes of acute rejection, the patient was in a stable CKD condition. The abrupt development of severe microscopic hematuria and renal dysfunction was observed approximately 2 weeks after the onset of a phlegmon in his right leg. An allograft biopsy showed prominent glomerular endocapillary proliferation on light microscopy, granular C3 deposition on immunofluorescent microscopy, and subepithelial electron-dense deposits on electron microscopy, suggesting IRGN accompanied by moderate interstitial fibrosis and tubular atrophy (IFTA). Positive glomerular staining for nephritis-associated plasmin receptor (NAPlr) and plasmin activity, which are biomarkers of bacterial IRGN, supported the diagnosis. Although the infection was completely cured with antibiotic therapy, renal dysfunction persisted. A re-biopsy of the allograft 2 months later revealed resolution of the glomerular endocapillary proliferation and negative staining for NAPlr/plasmin activity, with worsening IFTA. We showed, for the first time, the chronological changes in infiltrating cells and histological markers of IRGN in transplanted kidneys. Glomerular changes, including NAPlr/plasmin activity staining, almost disappeared after the cessation of infection, while interstitial changes continuously progressed, contributing to ESKD progression.

## 1. Introduction

The epidemiology of glomerulonephritis triggered by infection has changed dramatically over the past 30 years. The number of cases of post-infectious acute glomerulonephritis (PIAGN), which is mostly caused by streptococci and is common in children, has decreased, while the number of cases of nephritis in adults caused by various bacteria has increased. As infection in these cases is not cured at the onset of nephritis, it is now more widely referred to as infection-related glomerulonephritis (IRGN), including the conventional term PIAGN [1]. Some proportion of IRGN in adults has poor prognosis even after complete eradication of the causative infection, the detailed mechanism of which remains unclear [1,2]. IRGN in transplanted kidneys is extremely rare despite frequent opportunities for infection due to the immunosuppressed state; however, it is generally known to have a poor prognosis, the reasons for which are unknown [3,4,5,6,7,8,9,10,11]. 

Here, we describe a patient with typical IRGN that developed in the transplanted kidney due to a phlegmon in the lower leg; renal biopsies were performed serially over a duration of 2 months. Although the glomerular lesion was typical of IRGN and almost completely resolved within 2 months, the patient progressed to end-stage kidney disease (ESKD). As there is no established animal model for IRGN, this case is suggestive and valuable in terms of the mechanism of IRGN exacerbation. Therefore, we report this case with a detailed histological analysis, including histological staining for nephritis-associated plasmin receptor (NAPlr) and plasmin activity, and an immunohistochemical analysis of the infiltrating cells and myofibroblasts.

NAPlr was originally isolated from the cytoplasmic fraction of group A Streptococcus as a candidate nephritogenic protein for post-streptococcal acute glomerulonephritis (PSAGN), and glomerular deposition has been observed in most cases of acute-phase PSAGN [12,13]. Furthermore, the plasmin activity in glomeruli stained by in situ zymography using plasmin-sensitive substrates has been shown to be distributed in a manner similar to glomerular NAPlr deposition, suggesting that NAPlr deposited in glomeruli binds to plasmin and maintain their enzymatic activity in the glomeruli, thereby causing glomerular damage by degrading extracellular matrix proteins and activating and accumulating inflammatory cells [13]. Subsequent studies have also revealed that glomerular-positive staining for NAPlr and related plasmin activity is widely observed in patients with bacterial IRGN other than PSAGN. Therefore, glomerular-positive staining for NAPlr and plasmin activity has been suggested as a general diagnostic biomarker for bacterial IRGN [14].

An analysis of this case suggests that although the glomerular lesions, including diffuse endocapillary proliferation and positive staining for NAPlr/plasmin activity and alpha-smooth muscle actin (α-SMA: a marker for activated mesangial cells and myofibroblasts), were rapidly and almost completely diminished, inflammation and fibrosis of the tubulointerstitial lesions progressed and contributed to the progression to irreversible renal failure.

## 2. Case Report

A 47-year-old man who suffered from a phlegmon of his right lower leg and presented with sepsis and acute-on-chronic kidney injury was referred to our department. He was diagnosed with ESKD due to diabetic nephropathy at the age of 44 years, when he underwent living-donor kidney transplantation with his younger brother as the donor. Following surgery, the patient recovered well without hemodialysis. However, he experienced worsening kidney function due to several episodes of acute T cell-mediated rejection and chronic antibody-mediated rejection. After these episodes, he was in a stable chronic kidney disease (CKD) condition with a serum creatinine level of approximately 2.0 mg/dL on triple immunosuppressive therapy (prednisone, tacrolimus, and mycophenolate mofetil). Diabetes management was also relatively good, with his level of HbA1c at 5.8~7.5%.

However, one day, at age 47, he developed a phlegmon in his right lower leg, a severe condition suggestive of sepsis, and an acute exacerbation of CKD, which resulted in urgent hospitalization and treatment. Figure 1 summarizes the course of events after hospitalization. 

A physical examination upon admission revealed low blood pressure and swelling of the right foot. His urinary test showed 2+ protein but no erythrocytes on microscopic examination. The results of the blood examination were as follows: white blood cell count, 19.1 × 10^3^/µL; hemoglobin, 7.6 g/dL; platelet count, 169 × 10^3^/µL; serum creatinine, 5.74 mg/dL; urea nitrogen, 86.2 mg/dL; total protein/albumin (TP/Alb), 5.6/2.7 g/dL; blood sugar, 116 mg/dL; HbA1c, 7.5%; C-reactive protein, 27.4 mg/dL; and procalcitonin, 349 ng/mL. Antibodies for HBV, HCV, and HIV were negative. Chest radiography suggested exacerbation of the bilateral pleural effusion. Computed tomography revealed a normal kidney allograft size. The patient was diagnosed with sepsis due to a phlegmon in the right lower leg and treated with antibiotic therapy and continuous hemodiafiltration (CHDF). Despite repeated blood and wound culture examinations, the etiology of the infection remained unclear. On the fifth day of hospitalization, CHDF was discontinued because of an increase in urine output. However, approximately 2 weeks after the onset of the phlegmon, his serum creatinine level suddenly worsened, with massive hematuria. Simultaneously, a low level of serum complement C3 (27.4 mg/dL) and a mildly decreased CH50 level (22.3 U/mL) were observed. The anti-streptolysin O (ASO) titer and anti-streptokinase (ASK) levels were within normal ranges. We restarted hemodialysis (HD) and performed an allograft biopsy on the 21st day of hospitalization. 

Light microscopy showed diffuse and global endocapillary proliferation (Figure 2A), as well as crescents in 2 of the 18 glomeruli (Figure 2B). Moderate interstitial fibrosis and tubular atrophy (IFTA) was also observed (Figure 2C). C4d immunostaining was negative, and the pathology report indicated no evidence of rejection. Immunofluorescence staining revealed the isolated granular 2+ deposition of complement C3 on the glomerular capillary walls without the deposition of immunoglobulins or other complement components. Electron microscopy revealed large electron-dense deposits at the subepithelial site of the glomerulus (Figure 2D).

Furthermore, direct double immunofluorescence staining for C3 and NAPlr with DAPI nuclear staining and in situ zymography for plasmin activity showed that all of these were positive in the glomeruli. The distributions of NAPlr and C3 were different, while the NAPlr and plasmin activities were similar (Figure 3A–D).

Based on these findings, the patient was diagnosed with IRGN accompanied by a moderate IFTA. The total amount of immunosuppressive drugs was decreased considering septic conditions (Tacrolimus hydrate, from 3 mg/day to 1 mg/day; Mycophenolate mofetil, from 1.5 g/day to 1.0 g/day). No additional corticosteroids were administered to the patient. During the following 2 months, his hematuria tended to be mild and his serum C3 levels increased to normal levels, but his renal dysfunction did not improve and his urinary volume severely decreased. To determine the cause of the prolonged renal dysfunction, particularly the possible involvement of rejection, a follow-up allograft biopsy was performed on the 88th day of hospitalization (Figure 1). 

Light microscopic sections of the eight glomeruli showed little endocapillary proliferation (Figure 4A) and severe IFTA (Figure 4B). C4d immunostaining was negative, and the pathology report indicated no evidence of rejection. Immunofluorescence staining showed isolated granular 2+ depositions of C3, mainly in the mesangium and weakly along the glomerular capillary walls (Figure 4C). Neither IF staining for NAPlr (Figure 4D) nor in situ zymography for plasmin activity was positive in the glomeruli of the second biopsy (Figure 4E).

The infiltrating cells in the first and second allograft biopsy specimens were identified via IF staining using cell-specific antibodies (T cells: CD3; macrophages: CD68) or enzyme histochemical staining for neutrophils (chloroesterase staining). As shown in Figure 5, the first biopsy tissue showed marked cellular infiltration mainly within the glomerulus, consisting mainly of neutrophils (blue cells) and macrophages (red cells), with some T cells (green cells) mixed in. In contrast, the infiltrating cells in the tubulointerstitial regions were primarily macrophages and T cells, with a small infiltration of neutrophils. 

The second biopsy tissue showed marked cellular infiltration mainly within the tubulointerstitial regions, consisting mainly of T cells and macrophages, with some neutrophils mixed in. Glomerular infiltration was mostly resolved at this stage; however, some macrophages and T cells, with a small infiltration of neutrophils, were observed (Figure 6).

We further analyzed the changes in the expression of α-SMA in both allograft biopsies. Regarding the glomerular staining, α-SMA was globally and strongly positive in the mesangial cells in the first renal biopsy, but the staining almost completely disappeared in the second renal biopsy. Strong interstitial staining of α-SMA was similarly observed in both biopsies (Figure 7).

These findings indicated that interstitial fibrosis was exacerbated even after the resolution of glomerulonephritis, leading to ESKD. Eventually, the patient was required to maintain chronic hemodialysis because of graft dysfunction.

## 3. Discussion

Here, we present a case of IRGN that developed in a patient with CKD who had received a kidney transplant and had a creatinine level of approximately 2 mg/dL. Allograft biopsies were performed twice over the duration of 2 months. Although the glomerular change in the first biopsy was typical of acute-phase IRGN and was almost completely resolved by the cessation of infection, the interstitial changes in IFTA continuously progressed, which might have contributed to the progression to ESKD.

Despite the high risk of infectious complications due to immunosuppressive therapy for allograft rejection, IRGN cases in renal allografts are extremely rare. Moroni et al. reported only three cases of IRGN in 827 renal transplantations performed between 1983 and 1992 [3]. In addition, only 11 cases of IRGN in allograft kidneys were reported in 2014 [4]. Except for streptococcal organisms, the etiology of these infections has been diverse in most cases [5,6,7,8,9,10,11]. The etiology of the infection in this case was unclear despite frequent culture tests; however, the presence of infection in his right lower leg was definitely evident. The clinicopathological features of this case, that is, the development of massive hematuria and hypocomplementemia approximately 14 d after the onset of infection in the lower leg, the glomerular histology of diffuse endocapillary proliferative glomerulonephritis with massive glomerular infiltration of neutrophils, the strong IF staining for C3 along the capillary walls, and the existence of a subepithelial hump, strongly suggest the diagnosis of acute-phase IRGN. Furthermore, the second renal allograft biopsy obtained 2 months later showed prominent resolution of the glomerular proliferative changes and disappearance of NAPlr/plasmin activity staining, suggesting the resolution of IRGN. As described above, IRGN in transplanted kidney is extremely rare, and reports on the histological change in serial biopsies are further limited. To the best of our knowledge, only one report exists to date [11], making this case extremely valuable.

The reason for this paradoxical rarity of IRGN in transplanted kidneys remains unknown, but the following possibilities are possible. First, patients of renal transplantation may not be tested for urinalysis following their transplantation frequently enough, allowing microscopic hematuria to remain unnoticed. Second, the immunosuppressive status of the allograft recipients may inhibit the development of immune complexes and IRGN. Third, prompt administration of antibiotics to treat bacterial infections may prevent the onset of IRGN in transplant recipients.

In the present case, the histological trend of the glomerular lesions was similar to that in typical PSAGN cases, with almost complete healing over time. However, the histological trend of tubulointerstitial lesions was completely different from that of typical PSAGN cases. Indeed, the glomerular positive staining for NAPlr/plasmin activity, the prominent neutrophil and macrophage infiltration, and the strong mesangial-positive staining for α-SMA almost completely disappeared over 2 months. In contrast, the tubulointerstitial changes, such as the infiltration of T cells and macrophages, the positive staining for α-SMA (myofibroblast infiltration), and the interstitial fibrosis, were observed with persistence or exacerbation, which may have led to the development of the severe IFTA and the clinical progression to ESKD. Similarly, Nguyen et al. pointed out the importance of interstitial changes in allograft IRGN in ESKD progression [11]. The prognostic relevance of the tubulointerstitial infiltration of T cells and macrophages has also been reported in other forms of glomerulonephritis in native kidneys, such as IgA nephropathy [15].

The reason for the poor prognosis of IRGN in transplanted kidneys may also be attributed to the presence of certain background interstitial lesions from the transplantation and the tendency for interstitial damage to progress even after an improvement in glomerular inflammation. Indeed, in this patient, several episodes of rejection at the early stage after renal transplantation resulted in renal dysfunction with a serum creatinine level of 2 mg/dL and a histological change of IFTA, which may have been related to the poor final prognosis. In this regard, the staining nature of α-SMA is suggestive and is one of the reasons that we considered it as a possible mechanism of progression from IRGN to CKD in a recent review [16].

The prognosis of IRGN in older individuals is poor, with less than a quarter of patients fully recovering their kidney function [2]; however, the detailed mechanism remains unclear. Although this patient is relatively young, he has a background of diabetes and an immunocompromised condition due to the use of immunosuppressive drugs for transplantation. Thus, this case is very suggestive for understanding the pathogenesis of IRGN in the elderly. The poor prognosis of IRGN in older individuals may similarly be due to the presence of background chronic interstitial lesions and their continuous progression after the resolution of glomerular lesions.

In conclusion, we report a case of IRGN that developed in a renal allograft and observed its changes over time using serial biopsy tissues. A histological analysis of this case suggested that the presence of moderate background interstitial lesions in the transplanted kidney may have led to a poor renal prognosis because of the tendency for interstitial damage to progress even after the resolution of glomerular inflammation. This may also explain the poor prognosis of IRGN in older patients with moderate background interstitial lesions. Furthermore, nephrologists should be aware that renal dysfunction may not improve in IRGN of the transplanted kidney, independent of rejection, and renal biopsy is useful in this regard.

## Figures and Tables

**Figure 1 ijms-25-05095-f001:**
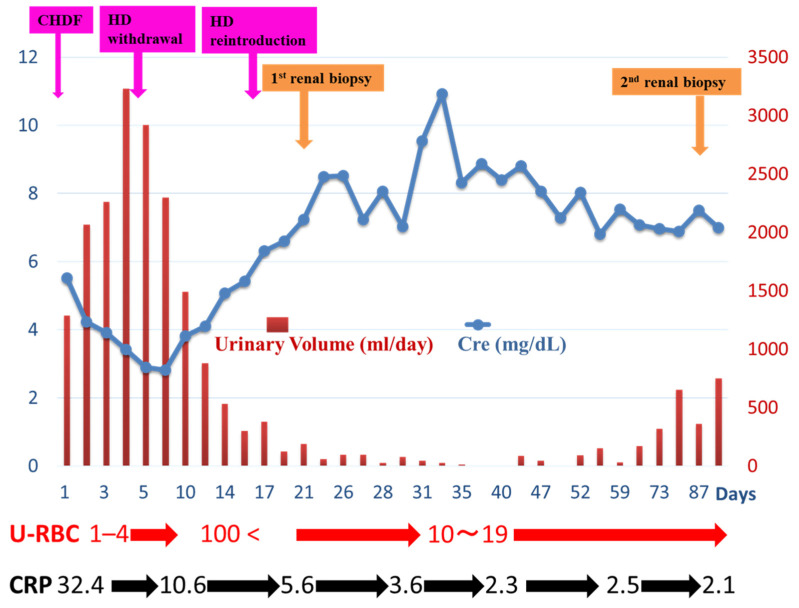
An overview of the clinical course after hospitalization. The data for serum creatinine (Cre), urinary volume, urinary RBC counts (U-RBC, cells/hpf), and CRP levels (mg/dL) over time are shown. The timing of the renal allograft biopsies and the initiation of continuous hemodiafiltration (CHDF), withdrawal from hemodialysis (HD), and reintroduction to HD are noted.

**Figure 2 ijms-25-05095-f002:**
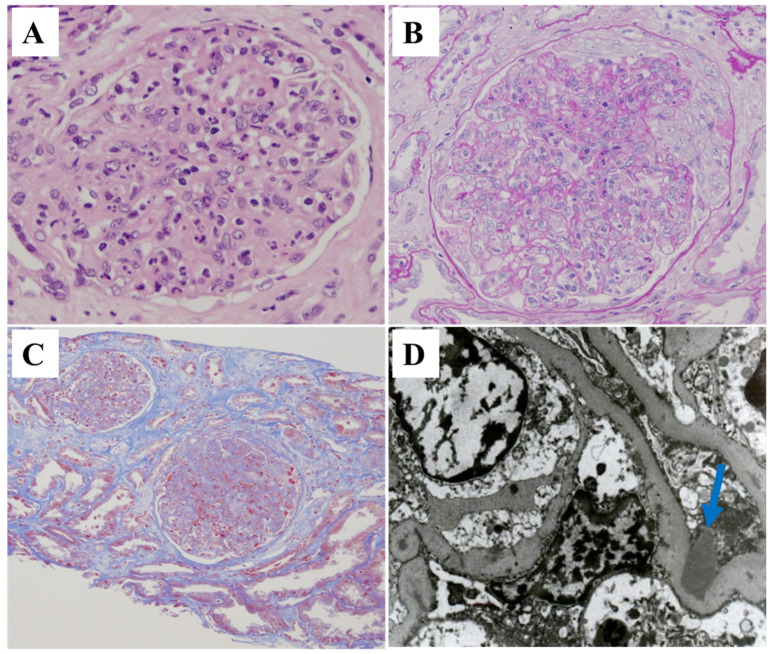
Histological features of the first renal allograft biopsy. (**A**) Diffuse global endocapillary proliferation with massive infiltration of neutrophils was observed by light microscopy (hematoxylin and eosin stain). (**B**) A cellular crescent with endocapillary proliferation was observed in some glomeruli (periodic acid silver-methenamine stain). (**C**) Moderate tubulointerstitial fibrosis was observed in the tubulointerstitial area (Masson trichrome stain). (**D**) Large electron-dense deposits were observed at the subepithelial site of the glomeruli (a so-called hump, indicated by the blue arrow). Original magnification: (**A**,**B**) 400× and (**C**) 100×.

**Figure 3 ijms-25-05095-f003:**
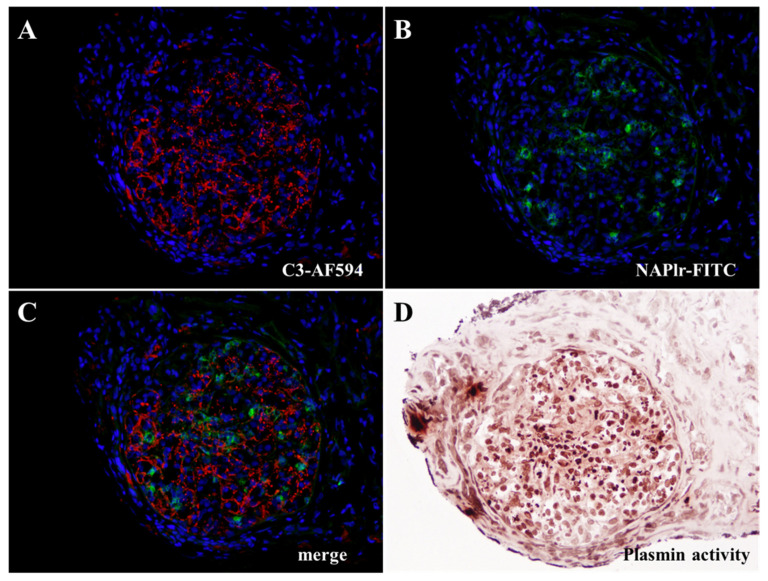
Representative photomicrographs of the histological staining for C3, nephritis-associated plasmin receptor (NAPlr), and plasmin activity in the glomeruli on the first biopsy specimen. (**A**–**C**) Double immunofluorescence (IF) staining for C3 (Alexa Fluor 594, red) and NAPlr (fluorescein isothiocyanate [FITC], green) with nuclear staining for DAPI (blue). Both NAPlr (**A**) and C3 (**B**) were positive in the glomeruli in different distributions, as shown in the merged image (**C**). (**D**) The plasmin activity assessed by in situ zymography on a serial section was found to be positive and had a similar distribution to the NAPlr staining in the glomeruli. Original magnification: (**A**–**D**) 200×.

**Figure 4 ijms-25-05095-f004:**
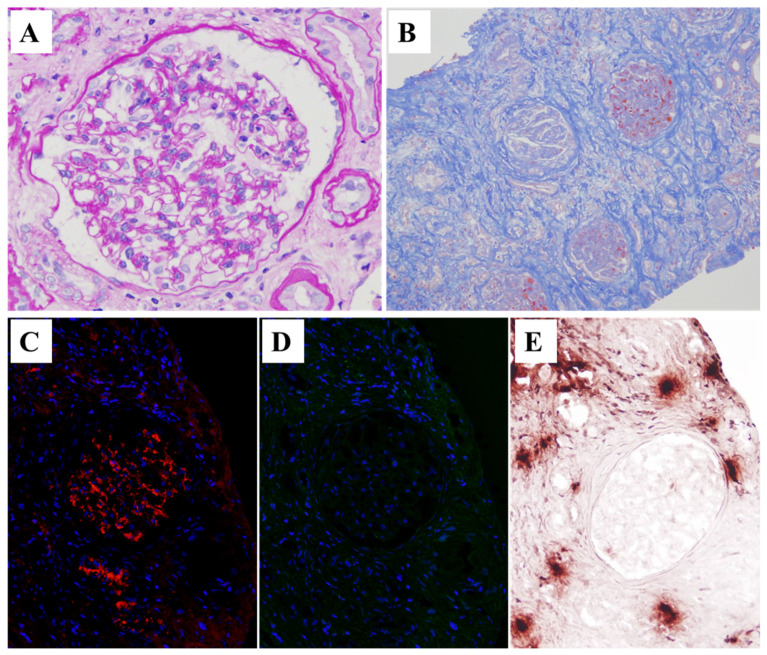
Histological features of the second renal allograft biopsy. (**A**) Endocapillary proliferation was mild in the glomerulus via light microscopy (periodic acid–Schiff stain). (**B**) Advanced severe fibrosis was observed in the tubulointerstitial areas (Masson trichrome stain). (**C**,**D**) Double immunofluorescence staining for C3 ((**C**) AlexaFluor594, red) and NAPlr ((**D**) fluorescein isothiocyanate, green) with nuclear staining for DAPI (blue). C3 showed 2+ staining mainly in the mesangial areas; NAPlr, however, was completely negative in the glomeruli. (**E**) In situ zymography for plasmin activity on a serial section also revealed a completely negative staining result in the glomeruli. Original magnification: (**A**) 400×, (**B**) 100×, and (**C**–**E**) 200×.

**Figure 5 ijms-25-05095-f005:**
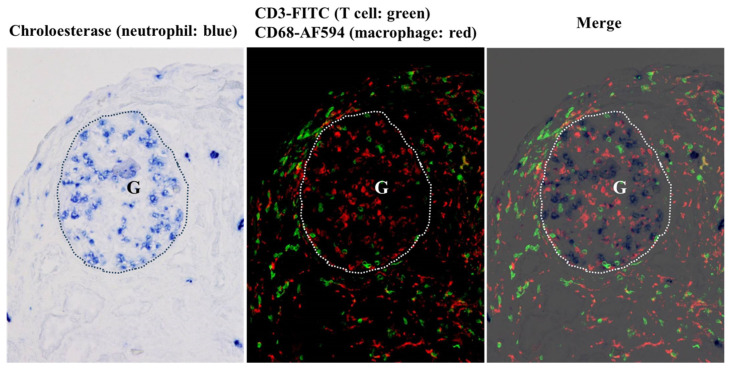
Representative photomicrographs of triple staining for neutrophils (chloroesterase staining, blue), T cells (CD3, fluorescein isothiocyanate [FITC], green), and macrophages (CD68, Alexa Fluor 594 [AF594], red) on a fresh frozen section of the first biopsy. The glomerulus is outlined with a dotted line and indicated by “G”. Neutrophils accumulated specifically in the glomeruli, while macrophages distributed diffusely in both the glomeruli and tubulointerstitium. T cell infiltration was also observed in both the glomeruli and tubulointerstitium. However, the density of the T cell infiltration was higher in the tubulointerstitium than in the glomeruli. Original magnifications: 200×.

**Figure 6 ijms-25-05095-f006:**
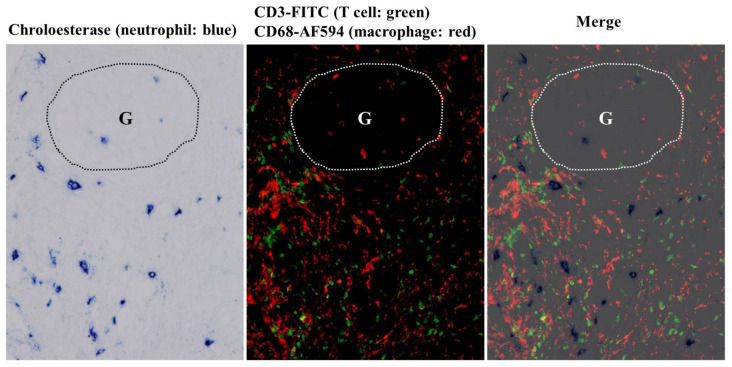
Representative photomicrographs of triple staining for neutrophils (chloroesterase staining, blue), T cells (CD3, fluorescein isothiocyanate [FITC], green), and macrophages (CD68, Alexa Fluor 594 [AF594], red) on a fresh frozen section of the second biopsy. The glomerulus is outlined with a dotted line and indicated by “G”. Compared to the first biopsy tissue, glomerular accumulation of the infiltrating cells, especially neutrophils, was markedly reduced. Tubulointerstitial infiltration of T cells and macrophages was still abundant. Original magnifications: 200×.

**Figure 7 ijms-25-05095-f007:**
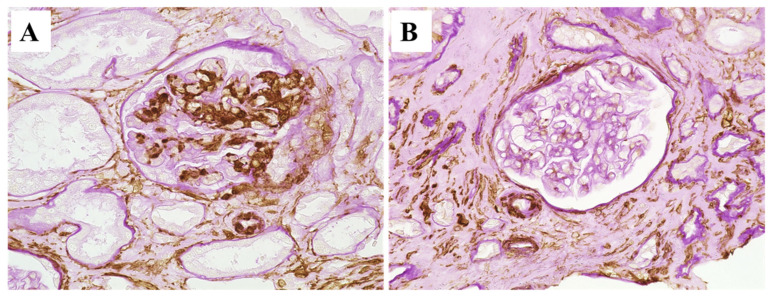
Representative immunoperoxidase staining for alpha-smooth muscle actin (α-SMA) counterstained with periodic acid–Schiff stain. Glomerular α-SMA staining was strongly positive on the mesangial cells (putative-activated mesangial cells) in the first biopsy (**A**), while its glomerular staining was pretty much weak in the second biopsy (**B**). Interstitial positive staining for α-SMA (putative myofibroblasts) was similarly strong in both biopsies.

## Data Availability

The data presented in this study are available from the corresponding author upon request. The data are not publicly available due to ethical and privacy limitations.

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
