# Peer review of "Chronological Changes in the Histology of Infection-Related Glomerulonephritis in Renal Allograft: A Case Report"

_ijms, 2024, doi:10.3390/ijms25105095_

Round 1
Reviewer 1 Report
Comments and Suggestions for Authors
In this report, kidney biopsies were performed over time in patient who developed infection-related glomerulonephritis (IRGN) after kidney transplantation, and the kidney tissue findings showed dramatic changes. You also performed multiple staining, and examined the significance of these findings. It was a very interesting and informative paper.
I think your paper would be better if you could write a little more about the details of the case.
Originally, adult IRGNs are known to have a poor prognosis, and even worse prognosis at older age. Risk factors seem to be patients with an immunocompromised background such as DM or cancer, and histopathological kidney tissue findings such as interstitial lesions. You have also written the above.
Although, this patient is young, he is immunocompromised state after kidney transplant, and had DM, which suggests that he has risk factors. On top of that, what I would like to know more is as follows. What was the patient’s DM status after the transplant? Dose acute rejection play a role in this poor prognosis? To what extent did you reduce the dose of immunosuppressive drugs when he developed IRGN?
In other words, it would be better to discuss whether this process is limited to transplant patients or the same for other IRGN patients.
There are many reports that IRGN has poor prognosis when there are interstitial lesions as histopathological finding. You point out the presence of T cell and macrophage in the interstitial area. Is there any relationship between this finding and poor prognosis? If there has not been a similar study before, I think this would be useful information, and should be considered carefully.
I believe this is a very important case report, so please consider it further.
Reviewer 2 Report
Comments and Suggestions for Authors
The authors present a clinical case of infection-related glomerulonephritis on a renal allograft. The authors perform two sequential allograft biopsies and thus document the chronological changes. Although there are some cases reported of infection-related glomerulonephritis after renal transplantation, it seems that this is the first case in which these chronological changes were evaluated. It is not clear why the second biopsy was performed. Was it only for scientific purposes (to assess this chronological evolution) and if so how were the risks vs. benefits balanced. Also, there were some interesting findings but I think, for the readers benefit, the practical importance of these findings should be explained.
Round 2
Reviewer 1 Report
Comments and Suggestions for Authors
Thank you for adopting my opinion.
By adding the patient’s condition, I understand that the patient’s renal function decline was not caused by acute rejection or diabetes mellitus.
I think we now have more insight into the deterioration of renal function due to IRGN.
Thank you for your hard work. I approve your publication.
Reviewer 2 Report
Comments and Suggestions for Authors
All the queries were responded in a satisfactory manner. I have no further questions or comments.